# Snoring Remediation with Oral Appliance Therapy Potentially Reverses Cognitive Impairment: An Intervention Controlled Pilot Study

**DOI:** 10.3390/geriatrics6040107

**Published:** 2021-11-01

**Authors:** Preetam Schramm, Namrata Das, Emet Schneiderman, Zohre German, Jason Hui, Duane Wilson, Jeffrey S. Spence, Pollyana Moura, Sandra B. Chapman

**Affiliations:** 1Department of Biomedical Sciences, Texas A&M University College of Dentistry, Dallas, TX 75246, USA; eschneiderman@tamu.edu (E.S.); german@tamu.edu (Z.G.); pollyanamoura@gmail.com (P.M.); 2Center for BrainHealth, University of Texas at Dallas, Dallas, TX 75235, USA; ndas4@partners.org (N.D.); jeffrey.spence@utdallas.edu (J.S.S.); schapman@utdallas.edu (S.B.C.); 3McLean Hospital, Harvard Medical School Affiliate, 115 Mill St, Belmont, MA 02478, USA; 4Department of Comprehensive Dentistry, Texas A&M University College of Dentistry, Dallas, TX 75246, USA; huidds@gmail.com; 5College of Dental Medicine, University of New England, Portland, ME 04103, USA; phillipdwilson@msn.com

**Keywords:** oral appliance therapy, Alzheimer’s disease, mild cognitive impairment, snoring, airway management

## Abstract

Respiration rate (RR) dynamics entrains brain neural networks. RR differences between mild cognitive impairment (MCI) and Alzheimer’s disease (AD) in response to oral appliance therapy (OAT) are unknown. This pilot study investigated if RR during stable sleep shows a relationship to pathological severity in subjects with MCI and AD who snore and if RR is influenced following stabilization of the upper airway using OAT. The study cohort was as follows: cognitively normal (CN; *n* = 14), MCI (*n* = 14) and AD (*n* = 9); and a sub-population receiving intervention, CN (*n* = 5), MCI (*n* = 7), AD (*n* = 6) subjects. The intervention used was an oral appliance plus a mouth shield (Tx). RR maximum (max) rate (breaths/minute) and RR fluctuation during 2116 stable sleep periods were measured. The Montreal cognitive assessment (MoCA) was administered before and after 4 weeks with Tx. Baseline data showed significantly higher RR fluctuation in CN vs. AD (*p* < 0.001) but not between CN vs. MCI (*p* = 0.668). Linear mixed model analysis indicated Tx effect (*p* = 0.008) for RR max. Tx after 4 weeks lowered the RR-max in MCI (*p* = 0.022) and AD (*p* < 0.001). Compared with AD RR max, CN (*p* < 0.001) and MCI (*p* < 0.001) were higher with Tx after 4 weeks. Some MCI and AD subjects improved executive and memory function after 4 weeks of Tx.

## 1. Introduction

Alzheimer’s disease (AD) accounts for up to 75% of all dementia cases. In the U.S.A., associated AD deaths increased by 71% between 2000 and 2013. AD is now ranked as the sixth leading cause of death in the U.S.A. [1]. Typically, AD respiratory dysfunction manifests as sleep disordered breathing (SDB) [2]. The neural network involved in sleep and respiration regulation is complex with known brainstem controlling nuclei. However, there is a growing body of evidence indicating that synchronization of these neural connections might be organized by the entry of air into the nostrils during slow respiration [3]. Furthermore, airflow through the nose is critical for respiratory entrainment of delta and theta rhythms involved in cortical excitability, synchronized activity within cellular networks and coordination of signaling for shaping sensory coding, memory and behavior [4,5]. Time frequency analysis of inspiratory nasal and mouth respiration showed that nasal breathing entrainment has greater spectral power between the olfactory cortex, amygdala and hippocampus regions, demonstrating a three-way directional interaction among the route of breathing, emotion and respiratory phase [4].

It is well known that emotional states, including stress, anxiety and grief, are linked to limbic system involvement. Emotional states can also modify the rate and depth of breathing. Animal studies show that respiration-locked slow-wave oscillations are amplitude modulated and associated with slow-wave sleep (SWS) [5,6]. These reports indicate that respiratory rhythms can modulate some slow-wave oscillatory patterns throughout the brain.

SWS is reduced, and sleep continuity is often fragmented in patients with SDB. SDB is abnormal breathing during sleep. It is often associated with repetitive upper-airway collapse. In mild forms, it manifests as snoring. More severe forms present as apneas, defined as an absence of breathing occurring for ≥10 s or hypopneas—shallow breathing associated with a 30% decrease in airflow for ≥10 s. Apneas and hypopneas are time-linked with oxygen desaturation (hypoxia), electroencephalogram arousals, hypercapnia, and increases in blood pressure, heart rate and tachypnea. 

SDB is a stressor that can cause two fundamental physiological and well-documented responses to the associated sleep deprivation: reduced alertness and cognitive performance [7,8,9,10]. The spectrum of SDB ranges from upper airway flow limitation and snoring to obstructive sleep apnea (OSA). Evidence indicates that the snoring SDB subtype presents with the highest hypercapnia state (112%), compared with stable breathing (108.4%) and OSA (105.8%) subtypes [11]. Hypercapnia presenting during snoring periods may be more important than hypoxia implication in neurocognitive impairment [12]. Cognitive impairment reversibility was shown in AD patients with mild to moderate OSA treated for 3 weeks with continuous positive airway pressure, resulting in improved executive functioning and psychomotor speed measured with the Mini Mental State Examination (MMSE) [13,14,15,16].

Furthermore, SDB-related brainstem tauopathy can be detected in postmortem brainstems younger than 20 years old [17], suggesting that this important public health issue starts earlier than its categorization as an ‘old age’ risk factor. Additional evidence of SDB-related brain pathology is provided from a report that showed that gray matter volume reductions were present in prefrontal, frontal, and mid-brain as well as in the brainstem through the midline pons into the medulla in 16 children (6.9 ± 2.8 years) with severe OSA [18]. Since nasal airflow and brainstem nuclei influence respiration, cognition and autonomic nervous system regulation of sleep–wake oscillations [19], it is likely that respiration activity during sleep might be a more suitable area for investigation of selective vulnerability in the AD process. Our hypotheses are based on observations that respiration rate (RR; breaths/minute, bpm) changes occurring during stable sleep periods might reflect neural integrity, functioning and differentiate cognitively normal (CN) from AD and mild cognitive impairment (MCI) pathophysiology.

We used an intervention (Tx) that included (1) a midline traction oral appliance that advances the mandible anteriorly to increase upper airway patency, and (2) a mouth shield, to facilitate nasal breathing. Tx allowed us to test three inter-related hypotheses about breathing, sleep and cognition. The first hypothesis was RR during stable sleep periods differs between CN, AD and MCI subjects who snore. Second, RR fluctuation would differ in all three groups in response to Tx after 4 weeks of use. The third hypothesis was that RR response to Tx would influence Montreal cognitive assessment (MoCA) [20] test scores. Stable sleep is dominated by stable respiration, elevated arousal thresholds and disconnect from olfactory cortex activity [19]. RR changes during this critical period has not been fully elucidated nor has any published research assessed whether RR differences occur between AD and MCI subjects or in response to Tx during sleep. 

## 2. Materials and Methods

### 2.1. Study Design

This study was designed as an interventional controlled pilot study (Clinical Trials.gov. ID: NCT03929302). Subject randomization was not considered due to the pilot study’s exploratory nature and small group sizes and because categorization or diagnoses of all subjects was confirmed during the screening process by neurocognitive testing, or in the case of AD subjects, based on a neurologist’s clinical diagnosis. The screening process and detailed inclusion and exclusion criteria are described in Appendix A. The Institutional Review Boards (IRBs) at the University of Texas Southwestern Medical Center (UTSW), the University of Texas at Dallas (UTD) and Texas A&M University College of Dentistry (TAMUCOD; IRB2018-0785) approved the study protocol. Informed consent was provided and signed by all eligible participants in approved guidelines with UTSW, UTD and TAMUCOD IRBs. A detailed description of the neurocognitive tests used was published [21]. 

### 2.2. Subjects

A total of 41 intention-to-treat subjects [CN (*n* = 15), MCI (*n* = 15), and mild AD (*n* = 11)] were considered for inclusion based on Alzheimer’s disease neuroimaging initiative criteria. The number of intention-to-treat subjects who discontinued the study due to inability to adhere to the protocol schedule of events or were lost to follow-up after initial assessments are as follows: CN (*n* = 10 (not included due to temporomandibular joint (TMJ) issue, *n* = 2), MCI (*n* = 8, not included due to prior OAT use, *n* = 1), and AD (*n* = 5 (not included due to TMJ issue = 1)). The intention-to-treat group baseline sleep recordings were collected and analyzed for variables of interest from 37 subjects (90.2%; CN = 14, MCI = 14, AD = 9). From the 37 subjects, a sub-population of 5 (5) CN controls, 7 (7) MCI and 6 (6) Alzheimer’s disease (total *n* = 18; 48.6%) completed neurocognitive tests, pre- (baseline, BL) and Tx assessment, including sleep recordings after 4 weeks. The sub-population demographic data are presented in Table 1. All subjects were invited to undergo neurocognitive tests, at-home sleep recordings and trial Tx for 4 weeks during sleep. Inclusion criteria required subjects to be between 50 and 85 years old, have a minimum of 12 years of education, be native English speakers, and be right-handed. A history and confirmation of snoring (≥5 snore events/hour), which is the accepted symptom of upper airway resistance during sleep and precursor to OSA, was required. All subjects completed the MMSE [22], Geriatric Depression Scale [23], and the Wide Range Achievement Test 4 [24] as part of the screening. All subjects passed visual and auditory test requirements. No one was excluded based on race or ethnicity. Subject data were de-identified. Dental clinicians and sleep specialist were blind to subject status. 

### 2.3. Neurocognitive Testing

Under a clinician’s supervision at Center for BrainHealth, a division of UTD, eligible participants completed a 4 h testing protocol. The measures assessed the individual’s cognitive domains of memory (episodic and short-term memory), executive function (working memory, inhibition, verbal fluency (language), and reasoning), and attention (visual attention and selective attention). All neurocognitive test data were collected, summarized and transferred to a database spreadsheet by ND. For this manuscript, we investigated the mean change in MoCA test scores collected prior to intervention initiation and after 4 weeks of using Tx. 

### 2.4. Dental Assessment and Oral Appliance Intervention

Standard clinical periodontal exams were performed at BL to assess oral health and teeth stability for supporting the oral appliance. The myTAP oral appliance (AMI, Carrollton, TX, USA) is a dentist-fitted, midline traction design oral appliance (OA) applied chair-side requiring no laboratory services. It utilizes an upper and lower tray made of thermacryl coupled at a single-point midline mechanism that allows adjustment by 0.3 mm incremental advancement. The OA is forward adjusted to a starting position of the lower jaw at 60% of maximum protrusion based on our published work [25]. The mouth shield uses a silicone sleeve fitted over the OA midline mechanism to retain its position in the oral vestibule, providing comfort with OA use while facilitating nasal breathing. 

Participants were instructed to use Tx for 4 weeks during sleep for >5 h/night. Subjects could advance the jaw 0.3 mm every 2 nights if snoring persisted (less frequently if needed to minimize transient discomfort) without causing discomfort at the TMJ, masticatory muscles or teeth that lasts more than 1 h after removing Tx in the morning. All dental assessment data were collected, summarized and transferred to a database spreadsheet by PS. 

### 2.5. Sleep Data Collection

Each participant and/or caregiver received instructions on how to self-apply the FDA-cleared SleepImage recorder (MyCardio, LLC, Denver, CO, USA) to collect home sleep recorded data. The recorder collects the electrocardiogram (ECG) signal, actigraphy, body position, snore vibration and ECG-derived respiration (EDR). The collected ECG data are automatically analyzed using up-sampled 600 Hz data with frequency- and time-based cardiopulmonary coupling (CPC) algorithms that divide sleep states into “stable sleep” (0.1–0.4 Hz) that is respiration dependent and “unstable sleep” (0.01–0.1 Hz), driven by heartrate variability [26]. The apnea hypopnea index was not measured in this study because the recorder does not collect nasal airflow data. 

The first eight stable sleep periods within each recording (study total = 2116 periods) having a duration of >10 min were used to obtain the RR at the period’s start (RR start) and maximum (RR max). Five recordings had <8 stable sleep periods meeting the >10 min criteria (BL: CN 2 (7 periods) and 1 (6 periods), AD 1 (6 periods); Tx: CN 1 (4 periods). The RR fluctuation from RR start to RR max was calculated, using ImageJ (version 1.52a) software and are reported here in pixel units. A minimum of 5 recorded hours without artifact was considered acceptable. All sleep data were collected, summarized and transferred to a database spreadsheet by PS.

### 2.6. Statistical Analysis

A linear mixed model with nested random effects parameters (restricted maximum likelihood estimates) and fixed effects parameters (maximum likelihood estimates) for each condition and condition differences (interaction effects) for each group was used. The analysis compared groups against CN, conditions and interactions. All standard errors used for the tests were functions of the restricted maximum likelihood estimates. 

Normally distributed variables were summarized with means and standard deviations. Analysis of variance (ANOVA) followed by Bonferroni-corrected Wilcoxon signed rank tests were used for testing differences among and between time points, respectively; Pearson correlation analysis was used to evaluate the relationships to RR measures. RStudio Statistical package (version 1.4.1103) was used for computations [27]. An alpha level of ≤0.05 was used. Since it is a small exploratory study, tendencies toward significant results of 0.05 < *p* ≤ 0.20 were also reported.

## 3. Results

There were no significant differences with regards to age, education and gender among the subjects (*n* = 18), who completed BL and Tx assessments. MMSE scores in the AD group were significantly lower than MCI (*p* = 0.013) and CN (*p* = 0.023) (Table 1).

### 3.1. Intention-to-Treat Group

ANOVA followed by post hoc Bonferroni-correction of intention-to-treat group baseline data (CN-MCI and CN-AD) showed: no significant RR start bpm differences between groups (MCI, *p* = 0.737; AD, *p* = 0.223); a tendency toward higher RR max in CN compared with MCI (*p* = 0.143); and AD (*p* = 0.073); significantly greater RR fluctuation in CN (46.89 ± 19.58 pixels), compared with AD (38.81 ± 18.52 pixels; *p* < 0.001) but not between CN and MCI (*p* = 0.668) (Table 2).

### 3.2. RR-Start of Stable Sleep Period and Response to Tx after 4 Weeks

At baseline, no significant differences were observed from the linear mixed model analysis within groups. Under Tx conditions, AD RR start was significantly lower, compared with CN (13.62 ± 3.19 vs. 14.52 ± 3.51 bpm; *p* < 0.001). The RR start Group x Tx interaction between AD (∆) and CN (∆) showed a tendency toward significance (*p* = 0.190) (Table 3).

### 3.3. RR Max during Stable Sleep Period and Response to Tx after 4 Weeks 

The linear mixed model analysis results indicated a Tx-BL (∆) effect (F (1109) = 7.337; *p* = 0.008). Within the group RR max, the comparison indicated a lower RR max (*p* = 0.005) with Tx after 4 weeks in the AD group, compared with BL. No significant BL RR max differences were observed, compared with Tx after 4 weeks in the CN and MCI groups. Comparisons between groups at BL showed no significant differences. Tx RR-max was lower in AD, compared with CN (*p* < 0.001) but not in MCI compared with CN (*p* = 0.738). The AD group change (∆) from the mean showed a tendency toward significance (*p* = 0.081), compared with CN. No significant change (∆) from the mean was observed between MCI and CN (*p* = 0.279) (Table 3).

### 3.4. RR Fluctuation during Stable Sleep Period: Baseline and Tx after 4 Weeks

Results of the linear mixed model analysis showed a significant group x Tx effect (F (2114.2) = 3.268; *p* = 0.042) and trended toward being different ((F (1114.1) = 3.123; *p* = 0.080) for a Tx-BL (∆) effect on RR fluctuation. With Tx, MCI RR fluctuation decreased significantly (50.62 ± 4.86 vs. 43.75 ± 4.87 pixels, *p* = 0.003). Tx RR fluctuation did not significantly change (∆) from baseline within CN and AD groups and between groups (MCI-CN, AD-CN). The change (∆) from mean RR fluctuation at baseline to Tx response was significantly greater in MCI (∆), compared with CN (∆) (−6. 87 ± 2.36 vs. 2.56 ± 2.84 pixels; *p* = 0.012). We observed a tendency toward greater change (∆) from mean RR fluctuation at baseline to Tx response in AD (∆), compared with CN (∆) (−3.73 ± 0.46 vs. 2.56 ± 2.84 pixels; *p* = 0.109). No statistical differences in change (∆) from mean RR-fluctuation were observed by group at baseline (MCI–CN, *p* = 0.258; AD-CN, *p* = 0.718) or at Tx (MCI-CN, *p* = 0.790; AD-CN, *p* = 0.692). (Table 3)

### 3.5. Snore Count Response to Tx after 4 Weeks

Tx after 4 weeks significantly decreased the snore count (BL, 278.74 ± 325.80 vs. Tx, 90.25 ± 143.97, snore count mean ± SD; *p* < 0.001) in all participants receiving OAT. Within each group, BL versus Tx showed significant decreases in snore count (CN-BL, 390.41 ± 346.48 vs. Tx, 149.36 ± 177.64; *p* = 0.036; MCI-BL, 267.61 ± 207.26 vs. Tx, 72.49 ± 74.80; *p* = 0.010; AD-BL, 163.56 ± 87.31 vs. Tx, 72.47 ± 71.15 snore count mean ± SD; *p* = 0.011).

### 3.6. Compliance to Tx

Based on the snore count (∆), one (14%) MCI and two (40%) AD subjects had < 20% difference after 4 weeks Tx from baseline, suggesting that Tx was not titrated (16.7%; 3/18 subjects). Our Tx compliance rate was 83.3% (15/18 subjects).

### 3.7. Montreal Cognitive Assessment (MoCA) Test Score—Response to Tx

MoCA test scores (normal range >26–30) decreased significantly in the CN group (28.60 ± 1.52 vs. 26.4 ± 1.52; *p* = 0.050), showed an increased tendency in the MCI group (25.86 ± 2.41 vs. 27.43 ± 1.90; *p* = 0.20), and did not change in the AD group (20.8 ± 3.11 vs. 20.6 ± 1.95; *p* = 0.642) with Tx after 4 weeks. Between group comparisons, BL showed higher MoCA test scores in CN compared with MCI (*p* = 0.049) and AD (*p* = 0.001). The MCI MoCA scores were higher than AD at BL (*p* = 0.009). Tx after 4 weeks increased MCI MoCA test scores such that they no longer differed significantly from CN at 4 weeks (*p* = 0.341). The AD group MoCA test scores remained lower, compared with those of CN and MCI (*p* < 0.001 and *p* < 0.001, respectively). At the individual level, after 4 weeks of Tx, 5 of 7 (71%) of MCI and 3 of 6 (50%) of AD subjects improved their MoCA test scores (Figure 1).

### 3.8. Intervention Related Adverse Events

No major adverse events related to Tx use and titration were reported.

## 4. Discussion

The results of this pilot study suggest that RR max and RR fluctuation during stable sleep periods demonstrate potential as biomarkers to differentiate CN from AD and MCI subjects comorbid with snoring. Furthermore, both variables detected an overall Tx response after 4 weeks in our sub-population receiving OAT, although the differences observed trended toward significance with RR fluctuation. These differences may be ascribable in part to snoring attenuation and nasal breathing facilitation with Tx. Future investigation will need to determine the impact each has on these outcome measures. 

RR measurements before and after Tx during stable sleep periods is a simple approach to determine whether RR differs between CN, AD and MCI subjects who snore. For targeted upper airway management efficacy in AD and MCI with OAT for snoring, the earliest symptom in sleep apnea remains to be elucidated in larger populations. The consequences of sleep apnea, cognitive and quality of life responses to CPAP were reported in adults with AD [13,14,15,16,18]. The traditional approach to assess AD using episodic brain scans are expensive, time-consuming and unable to capture the dynamic nature of AD processes. RR max and RR fluctuation utilized for tracking health status, using sleep as a ‘window of health’ may provide a dynamic view of disease processes. 

RR start did not differentiate the three groups and might be due to influences from faster frequencies dominating respiration modulation [5]. RR max during stable sleep periods could be the optimal time-linking of respiration-locked sensory activity to modulate delta oscillatory neuronal activity in the neocortex [5]. This potential link between respiration influences, cortical activity, and airflow through the nose was reported to entrain human limbic oscillations and modulation of cognitive function [4]. RR max could be a potential biomarker to identify different levels of cognitive decline, based on our findings from the intention-to-treat groups. Furthermore, in the treated sub-population, there was a marginally significant trend toward greater change in RR max in AD with Tx, compared with CN (*p* = 0.081). Overall, the Tx change in mean RR max demonstrated an order of AD > MCI > CN (Figure 2). If this order is replicated in a larger, adequately powered sample, RR max may be a tool to objectively differentiate AD and MCI neuropathological levels. Furthermore, after 4 weeks of Tx, RR fluctuation was significantly reduced (*p* = 0.012) in MCI, compared with CN, suggesting that Tx had slowed respiration delta frequency oscillations.

RRmax (∆ mean) with Tx was smallest in the CN group suggesting that homeostatic regulation of cardio-respiration dynamic equilibrium during sleep was conserved. The mean RR max (∆ mean) was almost two times greater in AD, compared with MCI. We speculate that this shift might represent some plasticity remaining within the neuronal network. 

Among the MCI group, the lower MoCA scores at BL compared with CN, increased in 71% (5/7) of MCI subjects and were not statistically different from CN after 4 weeks of Tx. The observed decrease in CN MoCA scores is within normal variation limits, and Tx remained >26, the cognitively normal cutoff for this instrument. The MCI increased MoCA score >26, which is noteworthy because this group was shown to consistently score below 26 [28]. It is very encouraging to observe cognitive improvement in MCI after 4 weeks with Tx. Future research will need to determine the amount of time required to achieve an improvement threshold in MCI. The AD group did not show improved MoCA scores although individually, 50% (3/6) of AD subjects had higher MoCA scores with Tx after 4 weeks. Longer periods of Tx use with follow-up cognitive testing might find that AD subjects need additional time to demonstrate a meaningful cognitive improvement response, if any. 

Compared with CN, the differences in RR during stable sleep in MCI and AD could be related to selective loss or dysfunction and allostasis among specific brainstem neuronal populations or subpopulations, which likely impacts physiological homeostasis. It is possible that the AD-related changes seen in brainstem nuclei are associated with neuronal dysfunction and increased rates of somatic morbidity and mortality [29,30]. The change in RR max with Tx was 48.5% in AD. Brainstem scans may help elucidate if this difference in RR between the two groups could be related to the degree of brainstem neuronal loss or dysfunction. 

The findings of our pilot study should be interpreted with caution, due to the small intention-to-treat sample, Tx sub-population size and absence of randomization. One of its strengths is its design that includes a CN control group. Another strength is that the dental clinicians’ and RR assessments were blind to subject grouping. To minimize the complexity and increase compliance of at-home sleep recordings in our study population, we did not collect nasal airflow, thoracic and abdominal respiratory effort and oximetry or invasive esophageal pressure measures to determine an apnea hypopnea index. We relied on a history of snoring that was confirmed to be associated with autonomic nervous system arousals (i.e., increases in heart rate) from baseline sleep recordings and support using this approach by a study of snoring children who had “normal” polysomnography findings yet presented with low cognitive function [31]. We did not conduct an interrater reliability test for subject diagnosis but relied on the diagnosis and experience of the referring clinician’s assessments of our subjects’ cognitive statuses. The baseline MoCA scores suggest that the subjects were clinically well characterized. The relationship between RR fluctuation, RR max and the Tx response and its clinical significance remains to be further investigated. In our statistical analysis, medications were not included as covariants, due to the small sample sizes. It is important to note here that all AD, but no MCI subjects, were prescribed anticholinergic medication. Previous experience with anticholinergic’s impact on sleep architecture suggests that the AD group should have faster rather than slower RR max [32,33]. Our work highlights the importance in approaching treatment alternatives for MCI and AD snorers through other perspectives, such as upper airway stabilization and nasal breathing facilitation, particularly because of the significant public health concerns associated with AD.

Nasal respiration entrains human limbic oscillations and modulates cognitive function [4]. When breathing is diverted from the nose to the mouth, oscillation power respiration peak dissipates [4]. In addition to organizing neuronal population activity across brain regions to orchestrate complex behaviors affiliated with orofacial sensation [33], breathing enhances memory retrieval [4]. This functional property may have been established early in the evolution of cerebral cortical circuitry when olfaction was a dominant sense and likely formed the tight connections between the olfactory cortex, hippocampus, neocortex and forebrain [34]. Neural networks increase or impede signal transmission in given frequencies and constitute the main framework for neural signal processing [35]. The degree of oscillation synchrony and amplitude vary continuously with several functions. For example, one purpose is to act as coding for brain functional activity (i.e., respiration-related slow oscillations, <1.5 Hz) [5]. Our observation of distinct RR max frequencies between MCI and AD suggests that the neural networks within each group are operating differently. The recent finding of Lucke et al. reporting an association between impaired cognition with increased respiratory rate and oxygen saturation <95% from wake vital signs lends support to our results [36]. 

Additionally, our findings are supported by a recent report that found a significantly slower decline in MMSE scores in AD patients with severe OSA at 3-year follow-up, who had been treated with CPAP [16]. The individual outcomes of improved cognitive impairment in 71% of MCI and 50% of AD snorers supports the 3-year follow-up study, but also adds to the literature by demonstrating that with Tx attenuation of snoring, a category of SDB that is considered a pre-mild OSA state with a prevalence of 16–89% in the general U.S.A. population [37] can improve cognitive functioning. Since snoring occurs early in the spectrum of OSA, preventative intervention measures should be implemented early and given clinical importance in order to mitigate progression to cognitive impairment. This study supports the use of OAT to attenuate snoring in MCI and AD subjects and demonstrates the potential to improve some executive and memory function after 4 weeks of use. Future well-powered long-term studies will be required to confirm if Tx sustains, continues to improve, delays or prevents MCI subjects from transitioning to AD.

## 5. Conclusions

Our pilot study findings showed after 4 weeks of Tx, some MCI and AD subjects improved their MoCA test scores. We also observed that snoring was significantly attenuated in all groups with Tx after 4 weeks. The salient contribution of our results adds to the growing body of evidence that emphasizes the importance in treating SDB and, in this case, snoring in these populations.

## Figures and Tables

**Figure 1 geriatrics-06-00107-f001:**
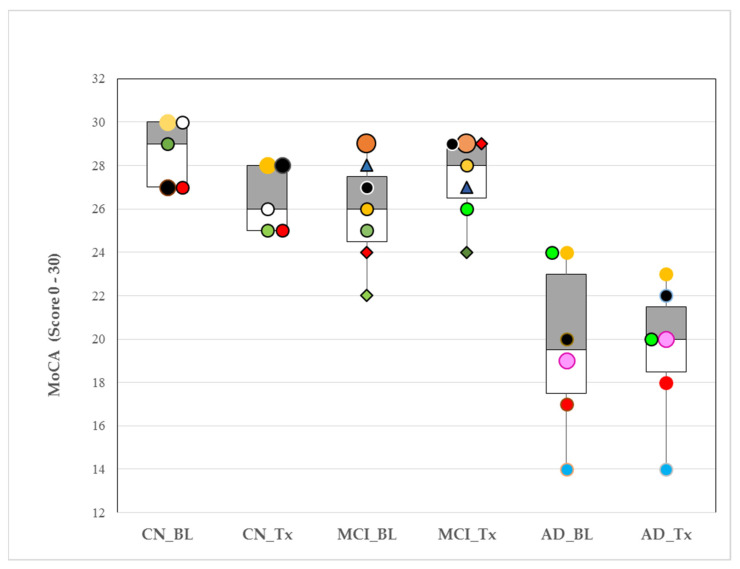
Box plots of medians and interquartile ranges (25th–75th percentiles) for Montreal cognitive assessment (MoCA) responses to oral appliance therapy (OAT) after 4 weeks in cognitively normal (CN), mild cognitive impairment (MCI) and Alzheimer’s disease (AD). Pre-OAT (baseline, BL); after 4 weeks (Tx, with OAT). *p*-values for within-group changes for CN, MCI and AD samples were 0.05, 0.20 and 0.642, respectively. MoCA normal range: >26–30. Colors and symbols identify individuals within group at baseline and their response to Tx. (Example: CN_BL ‘black circle’ subject (MoCA score = 27) increased their MoCA score to 28 with Tx).

**Figure 2 geriatrics-06-00107-f002:**
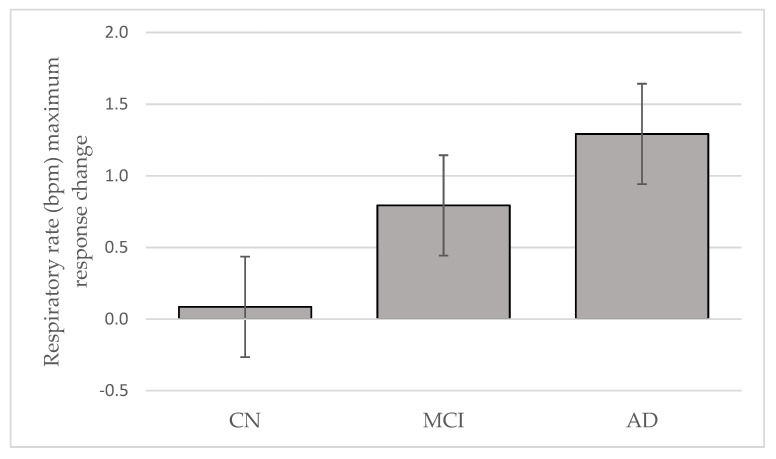
Respiratory rate maximum response change to oral appliance therapy after 4 weeks in cognitively normal (CN), mild cognitive impairment (MCI) and Alzheimer’s disease (AD) (Mean ± SE).

**Table 1 geriatrics-06-00107-t001:** Participants’ baseline characteristics receiving oral appliance therapy plus mouth shield.

	Group (*n* = 18)	CN (*n* = 5)	MCI (*n* = 7)	AD (*n* = 6)
Age (years) Mean ± SD (Range)	67.67 ± 6.68	65.60 ± 7.30 (56–75)	68.57 ± 7.55 (58–80)	68.33 ± 6.12 (62–75)
Gender (M/F)	(9/9)	(2/3)	(2/5)	(5/1)
Education (years) Mean ± SD (Range)	17.31 ± 2.83	18.0 ± 2.45 (16–22)	16.93 ± 3.68 (12–24)	17.17 ± 2.13 (14–19)
MMSE Mean ± SD (Range)	27.39 ± 2.07	29.20 ± 1.30 (27–30)	28.14 ± 1.35 (26–30)	25.00 ± 2.82 (21–28) *
Geriatric depression scale Mean ± SD (Range)	5.61 ± 4.74	3.20 ± 3.35 (0–8)	8.71 ± 5.47 (2–17)	4.00 ± 3.40 (0–9)

There were no significant differences among the groups except * AD Mini Mental Status Examination (MMSE) was lower than MCI (*p* = 0.013) and CN (*p* = 0.023); CN, cognitively normal; MCI, mild cognitive impairment; AD, Alzheimer’s disease.

**Table 2 geriatrics-06-00107-t002:** Intention-to-treat groups: ANOVA results of baseline Respiration rate-start, -maximum and -fluctuation between CN, MCI and AD.

		Mean ± SD		Contrasts (*p* Value)	
Variable	CN (*n* = 14)	MCI (*n* = 14)	AD (*n* = 9)	MCI-CN	AD-CN
RR start (bpm)	13.56 ± 2.13	13.44 ± 2.17	13.87 ± 2.24	0.737	0.223
RR max (bpm)	20.95 ± 3.18	20.46 ± 3.95	20.32 ± 3.22	0.143	0.073
RR fluctuation (pixels)	46.89 ± 19.58	45.77 ± 17.65	38.81 ± 18.52	0.668	<0.001

Abbreviations: CN, cognitively normal; MCI, mild cognitive impairment; AD, Alzheimer’s disease; Mean ± Standard Deviation); bpm, breaths per minute; RR, respiration rate.

**Table 3 geriatrics-06-00107-t003:** Comparison of intervention group respiration rate start, maximum and fluctuation during stable sleep periods.

			Contrasts						
	BL	Tx	Within Group	*p* Value	Between Group	*p* Value		Change (∆) from Mean	*p* Value
RR start (bpm) Tx-BL (∆)						BL	Tx	Tx-BL (∆)	0.763
CN	14.30 ± 3.40	14.52 ± 3.51	0.22 ± 0.31	0.968					
MCI	14.06 ± 5.53	14.34 ± 3.47	0.28 ± 0.25	0.156	MCI-CN	0.820	0.948	MCI(∆)-CN(∆)	0.875
AD	13.96 ± 2.18	13.62 ± 3.19	−0.34 ± 0.29	0.844	AD-CN	0.513	<0.001	AD(∆)-CN(∆)	0.190
RR max (bpm)								Tx-BL (∆)	0.008
CN	20.77 ± 3.40	20.62 ± 3.51	−0.15 ± 0.50	0.963					
MCI	21.88 ± 5.53	21.13 ± 3.47	−0.75 ± 0.41	0.269	MCI-CN	0.533	0.738	MCI(∆)-CN(∆)	0.279
AD	20.37 ± 2.18	18.86 ± 3.19	−1.51 ± 0.46	0.005	AD-CN	0.429	<0.001	AD(∆)-CN(∆)	0.081
RR fluctuation (pixels)								Tx-BL (∆)	0.080
CN	37.31 ± 5.71	39.87 ± 5.80	2.56 ± 2.84	0.963					
MCI	50.62 ± 4.86	43.75 ± 4.87	−6. 87 ± 2.36	0.003	MCI-CN	0.258	0.790	MCI(∆)-CN(∆)	0.012
AD	37.67 ± 5.22	33.94 ± 5.32	−3.73 ± 0.46	0.593	AD-CN	0.718	0.692	AD(∆)-CN(∆)	0.109

Abbreviations: CN (*n* = 5), cognitively normal; MCI (*n* = 7), mild cognitive impairment; AD (*n* = 6), Alzheimer’s disease; BL, baseline; Tx, oral appliance plus mouth shield; Change (∆), Change from Mean; RR, respiratory rate; Mean ± Standard Error.

## Data Availability

Data availability is inaccessible due to UTD Center for BrainHealth restrictions and the privacy and ethical agreement between the university and subjects. However, the summary data presented in this study are available on request from the corresponding author. The data are not publicly available due to privacy issues.

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
