# Peer review of "Snoring Remediation with Oral Appliance Therapy Potentially Reverses Cognitive Impairment: An Intervention Controlled Pilot Study"

_geriatrics, 2021, doi:10.3390/geriatrics6040107_

Round 1

Reviewer 1 Report

The manuscript title "Snoring remediation with oral appliance therapy potentially reverses cognitive impairment" is well written and scientifically sound. I have few suggestions listed below which the authors can include in the paper or can justify if they don't want to include them. Overall the paper is good and interesting to the readers.   1. As this topic deals with snoring it would be right to elaborate briefly on Obstructive sleep apnea. 2. Regarding data collection, please explain by whom and how the data collection procedure was done? 3. Explain briefly about inclusion and exclusion criteria 4. Explain why the periodontal examination was carried out? 5. Provide the questionnaire used and is it validated? 6. Explain various sleep disorders and their consequences in a detailed manner. 7. Explain the prevalence of sleep disorders in the study location and in the country. 8. Explain various present treatment procedures that are being practiced 9. Conclusion of the study is not well described  10. Explain the recommendations and study benefits in a brief manner.

Reviewer 2 Report

I have reviewed the manuscript “Snoring remediation with oral appliance therapy potentially re-verses cognitive impairment” submitted to “geriatrics” for publication. I found this work interesting and fit well with in the scope of this journal. The manuscript needs some major improvements; there are a few suggestions that authors may consider to improve it further:

The use of English language is reasonable, however, there are a number of punctuation and grammatical errors; that should be corrected and rephrased using academic English for a better flow of text for reader.

- Authors should mention the study design in the title to make it much clearer.

- Since this study is a controlled intervention, authors should explain why they didn’t use randomisation for this study?

- Few citations in the introduction are comparatively old. If probable, authors should replace these with latest one.

- Authors should mention limitations and future research directions in the discussion section.

Please provide IRB reference number if available.

- Authors should add conflict of interest heading before the references.

- While mentioning the years or months, authors should either add hyphen (eg. 2-months) or don’t add hyphen (eg. 2 months) throughout the document. The style should be consistent.

- Please carefully check the use of abbreviations in the abstract, main text, tables and figures to ensure the appropriateness.

In conclusions, please add suggestions based on the findings of the study.

Round 2

Reviewer 2 Report

Dear Authors,

Many thanks for the revision.